# The Characteristics of Natural Rubber Composites with Klason Lignin as a Green Reinforcing Filler: Thermal Stability, Mechanical and Dynamical Properties

**DOI:** 10.3390/polym13071109

**Published:** 2021-03-31

**Authors:** Jutharat Intapun, Thipsuda Rungruang, Sunisa Suchat, Banyat Cherdchim, Salim Hiziroglu

**Affiliations:** 1Faculty of Science and Industrial Technology, Surat Thani Campus, Prince of Songkla University, Surat Thani 84000, Thailand; namthip_1994rungruang@hotmail.com (T.R.); sunisa.su@psu.ac.th (S.S.); banyat.c@psu.ac.th (B.C.); 2Natural Resource Ecology and Management, Oklahoma State University, Stillwater, OH 74078, USA; salim.hiziroglu@okstate.edu

**Keywords:** Klason lignin, rubberwood, thermal stability, natural rubber composite

## Abstract

The objective of this work was to investigate the influences of Klason lignin as a filler on the thermal stability and properties of natural rubber composites. The modulus and tensile strength of stabilized vulcanizates were measured before and after thermo-oxidative aging. It was determined that lignin filled natural rubber had significantly enhanced thermo-oxidative aging and mechanical properties compared to those of controlled samples. The reinforcement effect of lignin increased stress with lignin loading but it decreased at 20 phr, suggesting that the reinforcement mechanism of lignin was via strain-induced crystallization. The composite samples with 10 phr filler loading had the highest mechanical properties as well as thermo-oxidative degradation resistance. Such a finding could be due to interactions between the Klason lignin filler and natural rubber matrix. Based on the findings in this work, the degradation temperature of Klason lignin occurred at 420 °C. The absorption peaks at wavenumbers 1192 and 1374 cm^−1^ indicated that C–O stretching vibrations of the syringyl and guaiacyl rings of hardwood lignin existed. It was also found that the Klason lignin–rubber composite containing 10 phr had the highest stress–strain, 100% modulus, and tensile strength, while lignin showed increasing aging resistance of the composite comparable with commercial antioxidant at 1.5 phr. It appears that Klason lignin from rubberwood could be used as a green antioxidant and alternative reinforcing filler and for high performance eco-friendly natural rubber biocomposites.

## 1. Introduction 

Natural rubber (*cis*-1,4-polyisoprene) is a variable raw material with excellent mechanical properties to be used for many applications, including automotive, construction and electronics industries [1]. However, an important issue is how to produce rubber composites with accepted stability. It is a known fact that natural rubber is highly unsaturated, resulting in poor resistance to oxidation [2]. It is also important that such degradation needs to be inhibited by adding some stabilizers. 

The biopolymer lignin is often considered to be a promising alternative green material. Lignin is an abundant biomass resource in aromatic structure with a low price which can be used as a renewable precursor for different value-added products [3]. Lignin is a three-dimensional amorphous natural polymer, which can be found in wood and other lignocellulosic materials at an approximate rate of 15–25% by weight [4]. It is a complex macromolecule based on three different phenylpropane units, namely, syringyl alcohol (S), guaiacyl alcohol (G) and *p*-hydroxyl alcohol (H) [5,6]. Lignin can be extracted by physical and/or chemical and biochemical methods [7,8]. Depending on the extraction method and the choice of plant source, the physical and chemical characteristics of lignin vary considerably [9,10]. Lignin can also be extracted from both soft and hardwoods including Rubberwood (*Hevea brasiliensis*), which is a light-colored medium-density tropical species. In general, the main types of lignin can be divided into lignosulfonate, kraft lignin, and organosol lignin. Recent advancements have been made on the use of ionic liquids to dissolve in the extraction of cellulose, lignin and lignocellulose with ionic liquids from the point of extraction efficiency and environmental friendliness [11]. Klason method, organosol lignin results in higher yield than alkaline or ionic liquid extraction [12]. Klason lignin is composed of a higher content of carbon, and a recent study reported that lignin can be considered as a viable carbon substrate for polyhydroxyalkanoates (PHA) as an important alternative product to replace synthetic plastics, chemical precursors and fuel-range hydrocarbons by applying various integrated processes [13]. Lignin is used as a precursor to the production of many types of aromatic compounds, activated carbon, phenol derivatives and antioxidants. The Klason method is also a simple, inexpensive and fast one, separating lignin as an insoluble material by depolymerizing/solubilizing cellulose and hemicellulose in 72% sulfuric acid (SA) followed by hydrolysis of the dissolved polysaccharides [14].

Lignin is also environmentally friendly as it has antioxidant and reinforcing capabilities [15]. A previous study has shown that the stabilization and antioxidant activity of lignin existed in natural rubber [16]. In addition, the kraft-lignin was used as a green alternative to silica in vulcanized rubber containing 20 phr lignin and 30 phr silica as a hybrid filler, showing optimal overall mechanical properties [17]. Various studies claimed that using hybrid technologies with carbon black [18], montmorillonite, and layered double hydroxides improved the mechanical properties of such compounds, with excellent reinforcing effect [19,20]. In addition, the utilization of modified lignin in styrene-butadiene rubber (SBR) had improved mechanical properties, such as tensile strength and tear strength [21]. Regarding the incorporation of softwood-lignin in natural and synthetic rubber, the increase in the mechanical properties is noteworthy; further, it is also possible to add plasticizers to the composite in order to enhance its overall mechanical properties by reducing the tendency of lignin particles to link together [22]. Currently, there is a great [16,18,23] research trend on variable mixing method to improve the reinforcement effect of lignin in rubber. Moreover, lignin in polyurethane has been developed as a substitute for petroleum-based polyurethane and used as a filler for the polyurethane industry [24,25]. It is a well-known fact that lignin–rubber composites have great potential as sustainable eco-friendly biomaterial.

However, there is very little information on Klason lignin from rubberwood to use as a green reinforcing filler and antioxidant for natural rubber. In this study, Klason lignin with a high carbon content and purity was extracted from rubberwood sawdust and characterized functional groups. Additionally, the overall influence of lignin loading level (0, 1.5, 5, 10, 15 and 20 phr) on cure characteristics, mechanical properties and the thermal stability of lignin/rubber composites was investigated. The thermal aging performance of lignin–rubber composites was compared with a commercial antioxidant (Butylated Hydroxyl Toluene, BHT). In addition, the lignin dispersion in the matrix and morphology were evaluated by employing scanning electron microscopy (SEM). The wet grip and rolling resistance were also discussed within the scope of dynamic properties. Based on the findings in this work, it is expected that developed composite samples with a green filler, namely Klason lignin, could have a potential to improve stabilization rubber based composites.

## 2. Experimental

### 2.1. Material 

The rubber wood sawdust (RWS) was obtained from the SPB panel company co. Ltd., Surat Thani, Thailand. The Standard Thailand Rubber 20 (STR20) was manufactured by Y.T. Rubber Co., LTD., Surat Thani, Thailand. The additives, namely zinc oxide (ZnO), *N*-tertbutyl-2-benzothiazole sulfenamide (CBS), Butylated Hydroxyl Toluene (BHT), stearic acid (SA), and sulfur curing agent, were purchased from Global Chemical Co., LTD., (Samut Prakarn, Thailand). Sulfuric acid used to extract lignin from RWS was supplied by Merck KGaA Darmstadt, Germany.

### 2.2. Klason Lignin Extraction Method and Properties

A rubberwood sample of 1.0 ± 0.1 g was placed in a 100 mL beaker before 15.0 mL of cold (10 to 15 °C) 72% sulfuric acid was added gradually in small increments while stirring and macerating the material with a glass rod. After the specimen was dispersed, the beaker was covered with a watch glass and kept in a bath at 20 ± 1 °C for 2 h, with frequent manual stirring. About 300 to 400 mL of water was added to a flask and the sample was transferred into this flask from the beaker. The material was then diluted to 3% with water before boiling for 4 h, maintaining constant volume either by using a reflux condenser or by frequent addition of hot water. The insoluble lignin was allowed to settle, keeping the flask inclined. In the next step, the lignin was quantitatively filtered using hot water flushing and washed free of acid with the hot water before it was dried in a crucible in an oven at a temperature of 105 ± 3 °C.

### 2.3. Homogenization Process and Particle Size Distribution Analysis

In the homogenization process, lignin was first dispersed in deionized water at 10% wt. lignin content. The total volume of the suspension was 200 mL, and homogenization was carried out using an Omni Sonic Ruptor 4000 Ultrasonic Homogenizer (Kennesaw, GA, USA) equipped with a small homogenizer head. The homogenization speed was 10,000 rpm. Particle size distribution analysis of a suspension in distilled water of lignin particles was carried out using a Fritsch Analysette-22 NanoTec laser particle size analyzer (Fritsch, Germany).

### 2.4. Preparation of NR/Lignin Composites

The lignin filled Natural rubber (NR) was compounded with the rubber additives displayed in Table 1 in a Brabender mixer (Zhanjiang, China). First, the NR was masticated in the mixer, and then the vulcanizing ingredients were mixed in in a conventional order, namely adding ZnO and SA followed by BHT or lignin, and CBS before sulfur was added last. The rotor speed was 60 rpm at a temperature of 60 °C for about 13 min. After cooling to room temperature, the rubber compounds were compressed by compression molding at a temperature of 160 °C for their respective optimum curing times, determined with a moving die rheometer (MDR).

The other ingredients in the composites were fixed at phr contents: ZnO 4.0, SA 1.0, accelerator CBS 1.6, and sulfur 1.4.

### 2.5. Fourier Transform Infrared (ATR-FTIR) Spectroscopy

Fourier Transform Infrared Spectroscopy (FTIR) measurements were carried out using PerkinElmer Spectrum (Perkin Elmer Inc., Waltham, MA, USA) with attenuated total reflectance (ATR) technique. This was qualitatively performed to characterize the functional groups and chemical characteristics of lignin. The resolution was 4 cm^−1^ and 32 scans were collected per recorded spectrum, in the wavenumber range of 4000 to 400 cm^−1^.

### 2.6. Mooney Viscosity and Cure Characterization and Bound Rubber Content of the Samples

The viscosities of the rubber compound were determined using a Mooney viscometer (MonTech, Tokyo, Japan) according to ISO 289-1. The curing characteristics of the compounds were determined at a temperature of 160 °C following the ISO 6502:1991 by using a Moving Die Rheometer (Montech, Tokyo, Japan).

### 2.7. Filler–Filler Interactions, Mechanical Properties and Thermal Aging Properties of the Samples

Properties of the compounded rubber were measured using RPA 2000 (Alpha Technologies, Akron, OH, USA). Strain sweeps were carried out with Rubber Process Analyzer (RPA2000, Alpha technologies Co.) at a temperature of 60 °C and 1 Hz frequency. The tensile tests were performed by a universal tensile testing machine (Tinius Olsen, model 10ST, Salfords, UK) at a temperature of 23 ± 2 °C. Dumbbell-shaped specimens according to ISO 527 (type 5A) were elongated with a cross-head speed of 200 mm/min. Aging properties of the rubber vulcanizates were also determined by characterizing the changes in the tensile properties after aging at a temperature of 70 °C for 3 days, according to ISO 188-1998, as compared with the unaged specimens. The retention of tensile properties was calculated as follows:(1)$$Retention\;\left(\%,\;\right)=\;\frac{\mathrm{Value}\;aged}{\mathrm{Value}\;unaged}\;\times \;100$$

### 2.8. Morphological Characterization, Thermal and Dynamic Properties of the Samples

Scanning electron microscopy (SEM), a Cambridge Stereoscan200 (FEI-Quanta 400, Hillsboro, OR, USA), was employed to visualize the morphological properties of the NR composites. The fresh fracture surfaces were first prepared by breaking the tensile test sample in liquid nitrogen. The energy dispersive X-ray (EDX) analyzer was also used to examine the chemical composition of the lignin. Two types of thermal properties of the samples were used in this work, namely thermogravimetric analysis (TGA), derivative thermogravimetric analysis, (DTG) and dynamic mechanical analysis (DMA). Thermogravimetric analysis (TGA) was performed by an STA 6000 (Perkin Elmer Inc., Norwalk, CT, USA) in the temperature range of 30 to 800 °C with a heating rate of 10 °C/min under nitrogen atmosphere. Dynamic mechanical analysis (DMA) was performed by a PerkinElmer DMA 8000 (Perkin Elmer Inc., Waltham, MA, USA) in the temperature range of −100 to 100 °C with a heating rate of 5 °C/min, and a frequency of 1 Hz. The storage and loss moduli together with tan δ were determined from the DMA thermograms.

## 3. Results and Discussion 

### 3.1. Properties of Extracted Klason Lignin

Klason lignin was prepared based on RWS by the Klason method, and the obtained content in the laboratory was about 21–23% by weight. The powder samples are black in color, and the particle size of the samples is depicted in Figure 1.

Figure 1 shows the particle size distribution of lignin from rubberwood. It can be clearly seen that the bimodal shape with two peaks was observed with average particle size (i.e., particle diameter). There are two major proportions of lignin at an average particle diameter of about 0.9 and 15.6 micron. The larger groups of aggregated Klason lignin particles may be due to the hydrophilic group character aggregation [26]. This range of the particle size of extracted Klason lignin in our laboratory preparation was similar to commercial hydrolytic lignin, which is a heterogeneous product of wood processing with acid. Particle size is an important factor for achieving a good interaction between the filler and polymer matrix. The reinforcing ability of lignin for rubber depends the particle size of the lignin and the interfacial interactions between the lignin and the rubber matrix [27]. Prepared Klason lignin had a particle size suitable for a reinforcing filler and able to incorporate to rubber. 

The content of Klason lignin from rubberwood in this study was about 21–24.6%, which is the initial amount for the rubber composite with a specific gravity value of 1.604 g/cm^3^. SEM/EDX analysis also provided a chemical composition of the lignin, as illustrated in Figure 2.

Lignin is the largest biomass component with aromatic properties and a high carbon content, which can be considered the most attractive sustainable precursor for carbonaceous materials, or of filler for rubber composites. Figure 2 shows SEM-EDX results of lignin from rubberwood. It can be seen that the chemical components in lignin consist of mainly carbon and oxygen, matching the natural lignin structure with a high weight percentage of carbon and oxygen, 45.54% and 54.46% by weight respectively. This type of lignin is very purified, no sulfur and has a high content of carbon, so that considered as a reinforcing filler for natural rubber, it would improve the thermal resistance of the rubber composite by the hydroxyl group of lignin.

### 3.2. Fourier Transform Infrared (FTIR) Spectroscopy

Fourier Transform Infrared Spectroscopy was used to investigate the chemical structure of the extracted lignin, as shown in Figure 3a, and the functional groups are displayed in Table 2. 

A broad characteristic peak was present in all the FTIR spectra, in the wavenumber range of 3396–3406 cm^−1^. This could be attributed to the stretching vibrations of hydroxyl groups in lignin molecules. The absorption peaks at approximately 2923 cm^−1^ and 2849 cm^−1^ were also observed, which correspond to stretching vibrations of C–H in methyl and methylene groups, respectively. Furthermore, the peaks at wavenumbers of 1704 cm^−1^ and 1614 cm^−1^ were found, as they are assigned to the stretching vibrations of C=O in phenolic rings of lignin [28]. The peak at wavenumber 1374 cm^−1^ is consistent with C–O stretching vibrations in syringyl rings, and 1192 cm^−1^ corresponds to C–O stretching vibrations of the guaiacyl rings [29]. Moreover, the absorption peak at wavenumber 1035 cm^−1^ corresponds to C–H in guaiacyl rings with C–O from primary alcohol, and 1112 cm^–1^ was assigned to the stretching of the C–H of syringyl rings. They are related to stretching vibrations of C–H bonds [11]. Such results correspond to Figure 3a, as the absorption peaks at wavenumbers of 778 and 619 cm^−1^ were observed. Similar results were also observed in two previous studies [30,31]. Therefore, it could be concluded that the extracted lignin from rubberwood contains both quaiacyl (G) and syringyl (S) and hydroxyl phenyl (H) units that can be attributed to C–O in the primary alcohol, as the proposed molecular structure of syringyl group in the lignin molecule is shown in Figure 4. It was found that the differences between the cases were in the intensities of the characteristic peaks as shown in Figure 3b. The strong peaks with the highest intensities, at 2963, 2921 and 2850 cm^−1^, correspond to the stretching vibrations of CH_3_, CH_2_, and CH groups in *cis*-1,4-polyisoprene macromolecules. However, the lignin particles also caused a small shift in this wavenumber range (Figure 3c). The intensive peak at 1539 cm^−1^ can be attributed to the symmetric aromatic skeletal vibrations in lignin macromolecules, C=C and C–H. The peak at 1448 cm^−1^ was assigned to deformations of methyl groups C–H, and is related to the lignin aromatic rings, and also to the natural rubber chains. The band near 1371 cm^–1^ could be attributed to the asymmetric vibrations of methyl C–H groups in natural rubber. The increasing intensity peaks at 1039 cm^−1^ were attributed to in-plane stretching of the aromatic C–O group in lignin [32]. This might be due to hydrogen bonding between the C–O group of lignin and the hydrophilic group of non-rubber content including protein and lipid in natural rubber [33]. The peak at 834 cm^−1^ is related to the vibrations of C–H groups in the rubber chains. The small shifts found in many wavenumbers indicate some interactions of C–H groups of aromatic of lignin and rubber, and the proposed hypothesis is shown in Figure 4.

### 3.3. Mooney Viscosity

The Mooney viscosity of rubber compounds distributed to interaction between the filler and rubber matrix. The high Mooney viscosity for compounds indicates that there is a high restriction on the molecular motion of the macromolecules, while low Mooney viscosity indicates that it could be easily processed [28].

Table 3 summarizes the Mooney viscosities of various NR/lignin compounds. It is clear that the Mooney viscosity decreased with the increase in the lignin loadings since lignin has a soft structure, causing a plasticizing effect in the rubber compounds. Therefore, it could increase the chain mobility, improving the overall flow behavior of rubber compounds [23]. This provides beneficial processing efficiency of the rubber compounds to promote the ease of natural rubber processing. The lower Mooney viscosity of lignin-filled natural rubber indicated that it could be processed more easily than unfilled-lignin natural rubber compounds.

### 3.4. Curing Characteristics of the Samples

The curing characteristics of natural rubber/lignin compounds are shown in Figure 5 and their data are summarized in Table 4. It can be seen that the cure time, tc90, and the scorch time, t_S_2, of lignin/NR compound slightly increased with the increase in the lignin loading, higher than those without lignin and with BHT addition excepted at 20 phr. Such a finding could be related to high levels of lignin affecting the efficiency of crosslinks in the vulcanizates and delaying the vulcanization of the NR composite. The vulcanization was delayed by the OH-containing lignin. Lignin also has a radical scavenging effect because it contains hindered phenols. The minimum torque, maximum torque, and delta torque of the 1.5 phr lignin-filled NR composites were insignificantly lower than those of the control (BHT) (Figure 5). The trend of maximum torque showed that 1.5 phr of lignin loading did not contribute to the increased shear modulus of the natural rubber composite during vulcanization. However, the minimum torque (ML), maximum torque (MH) and delta torque of lignin filled-NR composite increased significantly with increasing lignin content. ML is mainly associated with the physical crosslinking between the lignin and the rubber matrix before vulcanization. With the increase in lignin content, the rubber composite trends a higher value of ML, suggesting stronger physical crosslinking between the lignin and the rubber matrix. A similar trend was also observed for delta torque and crosslink density. This could be due to the higher crosslink density in lignin-filled NR compounds and the formation of a filler–rubber network between the lignin and rubber matrix. This would enhance the restriction of the mobility of rubber chains and would significantly reinforce the strength of the rubber composite [32].

### 3.5. Dependence of Shear Modulus (G’) on Strain

Strain sweeps at constant frequency and temperature were carried out with a Rubber Process Analyzer (RPA) to identify the dynamic properties of NR/lignin compounds with different lignin loadings. As can be observed from Figure 6, at above certain strain amplitudes the shear modulus (G’) decreased rapidly in all cases. This could be explained by the Payne effect that is associated with the destruction of filler networks, along with other deformation induced changes in the microstructure [17]. It can also be noted that the shear modulus (G’) of rubber decreased with the increase in the strain due to the Payne effect, which can be attributed to the fact that the shear deformation destroyed some filler network structures. It is clear that the NR with lignin loadings 1.5 and 5 resulted in low shear moduli and modulus difference (∆G’), indicating little filler–filler interactions and good filler dispersion in rubber. At 10 phr lignin, ∆G’ was noticeably higher, indicating the formation of lignin agglomeration in the natural rubber matrix, which is mainly caused by hydrogen bonding, leading to the formation of the filler network in the rubber matrix [28]. Therefore, it could be concluded that the shear modulus, modulus difference (∆G’) and the Payne effect increased with increasing strain amplitudes of the dynamic test in both cured and uncured NR/lignin compounds. However, the vulcanized rubber/lignin composite had an increase in the shear modulus with lignin for cured NR/lignin compounds, which could be due to the networks restricting the mobility of rubber chains which increase in the crosslinking of the rubber composite after curing process. 

### 3.6. Morphological Properties of the Samples

SEM imaging enables the assessment of the morphology of filler–rubber composites in the form of lignin distribution. Figure 7 shows the fresh fracture surfaces of the lignin-filled natural rubber composites. Lignin content at 10phr gave homogeneous filler dispersion with small particle sizes, and the fracture surface had small holes. Compared to 15 phr lignin-filled rubber, lignin content at 10 phr showed less agglomerated lignin than 15 phr. Such a finding indicates the full dispersion of lignin 10 phr in the rubber matrix and interfacial adhesive between lignin and rubber, matching the improvement with the mechanical properties including 100%, 300% modulus, toughness and tensile strength. The composites with 1.5 and 5 phr lignin had comparatively rougher surfaces and large holes, indicating poorer dispersion and interaction of filler to the matrix. The higher filler loadings may have induced agglomeration, which may be due to hydrophilic groups including phenolic hydroxyl, carbonyl of lignin, and corroborated to the decreasing interaction between rubber molecules and lignin [34].

### 3.7. Mechanical Properties of the Samples

Tensile stress–strain curves of lignin-filled NR containing 1.5, 5, 10, 15 and 20 phr filler along with the unfilled sample are illustrated in Figure 8. The reinforcement effect of lignin increased stress with lignin loading, but a decrease was again observed at 20 phr, suggesting that the reinforcement mechanism of lignin was via strain-induced crystallization [35]. The filler–rubber interactions in the composite with 10 phr are facilitated by the good dispersion and interaction of lignin in the rubber matrix, and this case also had the highest 100% modulus (Figure 9a) and tensile strength (Figure 10). As confirmed by SEM micrographs, NR/lignin at 10 phr had a much better distribution of lignin within the NR matrix compared to the non-filled composite and added above 10 phr (Figure 7).

Toughness is defined as the total energy absorption during extension, and equals the area under the stress–strain curve. Figure 8b shows the changes in toughness values with the addition of lignin. Toughness first increased with the addition of lignin but then decreased beyond 10 phr loading; therefore, 10 phr gave the maximal toughness of rubber film [33]. The stress–strain properties and toughness of Klason lignin/NR increased about 30% and 20% as compared to those of unfilled samples, respectively.

Figure 9 shows the 100% and 300% moduli of the lignin-filled rubber composite. The modulus of the samples increased with filler loading as a result of the interfacial adhesion of rubber and Klason lignin in the NR matrix. It can also be seen that the 100% and 300% moduli were maximal with 10phr filler loading in accordance with the SEM results, revealing good dispersion in the NR composite, and decreased with higher 10 phr lignin loading. Higher loadings resulted in lower values of moduli, possibly due to agglomeration and poorer filler dispersion.

It seems that lignin enhanced the strength properties of the rubber composite after it was aged at 70 °C for seven days, which is higher than using BHT antioxidant. This might be due to the lignin providing protection against rubber deterioration by heat, as demonstrated by the comparison of the moduli and tensile strengths before and after aging for the rubber composite. It is clear that the NR composite with 10 phr lignin gave the highest 100% modulus, 300% modulus and improved retention after aging. Apparently, the propylphenol structure of syringyl lignin with hydroxyl and methoxyl groups (as its molecular structure shown in Figure 4) in rubberwood confers protection against thermally induced aging.

Generally, natural rubber without antioxidants has a high tensile strength because it crystallizes when it is stretched. The lignin filler tended to improve the tensile strength (Figure 10), as Klason lignin acted as a reinforcing filler resulting in good dispersion in the rubber matrix. However, with a lignin content beyond 10 phr, the tensile strength of the samples decreased corresponds to a smaller of 100% Modulus and toughness. 

Natural rubber has unsaturated C=C bonds that can be degraded by heat and oxygen. Oxidation can cut unsaturated chains and bonds linked by sulfur. As a result, the tensile strength of the sample is significantly reduced after aging. The additive commercial BHT improved resistance to such degradation by acting as an antioxidant, with its phenol groups protecting against free radicals. It appears that the resultant group of phenolic hydroxyls can act as a stabilizer of oxidative reactions [36].

### 3.8. Thermogravimetric Analysis (TGA)

The thermal stability investigation (TGA and DTG thermograms) of Klason lignin and NR/lignin composite with a heating rate of 10 °C/min is shown in Figure 11 and Figure 12, respectively. In the thermogravimetric curve, the thermal degradation of lignin proceeded over a wide temperature range from approximately 100 °C to 800 °C.

It appears that the thermal degradation process of lignin took place within a temperature range from approximately 220 °C to 800 °C. This can be explained by the fact that lignin contains many aromatic rings with various types of chemical bonds and functional groups [37]. In Figure 11, in the first degradation stage over 30–100 °C the weight loss was less than 4%, which can be mainly attributed to the loss of moisture. After the first stage of weight loss, the degradation was slower over 130–220 °C, indicating that the main degradation of lignin took place around 220 °C, as shown in the insert at the top of Figure 11 which is the second degradation stage over 220–550 °C. In Figure 11 at the bottom left insert, the DTG peak that indicates the degradation temperature of lignin occurs at 420 °C. In a past study, it was determined that the Klason lignin from rubberwood showed a high decomposition temperature from hardwood and a higher thermo stability than the lignin from Eucalyptus (378 °C) [38], which could have contributed to the advantage of Klason lignin for preparing NR vulcanizates with higher degradation resistance properties.

Figure 12 depicts the TGA thermograms of natural rubber/lignin composites with various lignin loadings with their weight losses at various temperatures, as summarized in Table 5. The degradation temperature at 50% weight loss (T_50_) is considered to be a thermal stability index for rubber composites [37]. The results showed that the thermogravimetry of thermal resistance had two distinct temperature ranges, between 250–400 °C and 400–500 °C, indicating the deterioration of the natural rubber and the degradation of the components in the compound, respectively. The degradation resistance of NR composites increased with the incorporation of lignin, as displayed in Table 5. The addition of 1.5 phr lignin was a shift in the TGA peak (T_50_) to higher temperatures from 357.90 °C to 383.00 °C and higher than the use of BHT (377.67 °C), indicating the improved thermal stability of lignin-filled vulcanized rubber. It seems that lignin clearly improved the resistance to thermal decomposition as seen in the increasing T_50_, indicating a good heat resistance. This is attributed to the anti-oxidative effects of lignin via its phenolic hydroxyl groups that prevent oxidation.

### 3.9. Dynamic Mechanical Analysis (DMA) of the Samples

The storage modulus (E’) can be regarded as the elastic modulus of a rubber composite, and the loss tangent (E’’/E’) is related to the energy dissipation as heat [39]. The E’ values of NR/lignin composites with loadings 0, 1.5, 10 phr and 1.5 phr of BHT are shown in Figure 13a. It is clear that the storage modulus increased with lignin loadings and higher than the BHT addition, apparently because lignin hindered the molecular mobility of rubber, making the composites have less flexibility. It is noted that the E′ indicates the stiffness of the rubber composites, and its increase was attributed to enhanced lignin–rubber interactions. This correlates with the SEM results, confirming homogeneous distribution in the rubber matrix (Figure 7).

The tan δ of rubber composites is the ratio of the energy loss/energy stored (E’’/E’). The values of tan δ at 0 °C and 60 °C are associated with the wet grip property and rolling resistance of a rubber composite used in a car tire, respectively [40]. As shown in Figure 13b, the tan δ (60 °C) of lignin-filled NR lignin (0, 1.5 and 10 phr) and BHT addition was 0.03, 0.04, 0.04 and 0.15, respectively. The lower tan δ of the lignin-filled NR composite suggests a higher rolling resistance than BHT incorporation. The tan δ at 0 °C of lignin-filled NR (0, 1.5 and 10 phr) was 0.05, 0.07 and 0.06, respectively. These increasing tan δ values indicated an improvement in antiskid properties in wet conditions. In a word, the Klason lignin-filled NR composite improved wet grip properties and did not deteriorate the rolling resistance relative to unfilled rubber composites. This eco-friendly filler (Klason lignin) from rubber wood dust exhibited reinforcing properties, good antioxidant and improved dynamic properties for use as a green filler for natural rubber composites.

In further studies, the treatment method of Klason lignin by modifying process or using a coupling agent in order to increase interaction with the natural rubber matrix would be considered to investigate how to have a better understanding of its behavior. It is expected that the modification and treatment would also give the opportunity to use Klason lignin in more applications. Consequently, the combination of coupling and modification would benefit higher filler–rubber interaction.

## 4. Conclusions

In this study, a detailed characterization of structural Klason lignin and its thermal degradation was investigated. The antioxidant properties of Klason lignin as a reinforcing filler in natural rubber composites were also evaluated within the perspective of this study. It was determined that Klason lignin from rubber wood comprised both guaiacyl and syringyl structure with a high content of carbon and oxygen, which can be interacted with the non-rubber content of natural rubber. It can be concluded that lignin increased the scorch time, improving processibility, and also increased the delta torque, modulus levels of 100% and 300% and tensile strength of vulcanized rubber. The antioxidant properties of Klason lignin were found to be superior to the commercial antioxidant additive BHT. It appears that the 10 phr lignin loading was near optimal, resulting in the highest mechanical properties and aging resistance of the composite samples. The SEM analysis showed a homogeneous distribution of lignin particles in the rubber matrix, which improved the toughness property of the samples. DMA indicated that Klason lignin/NR had high wet grip properties, revealing that the samples had low rolling resistance, which could potentially make them suitable for green tire products. Moreover, the use of lignin as a filler not only facilitated the processing, but also provided environmentally friendly value-added rubber based composite products.

## Figures and Tables

**Figure 1 polymers-13-01109-f001:**
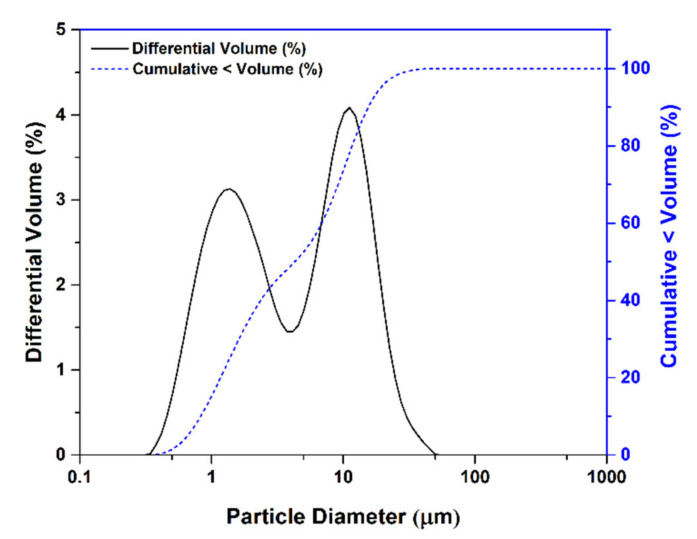
Particle size distribution of lignin.

**Figure 2 polymers-13-01109-f002:**
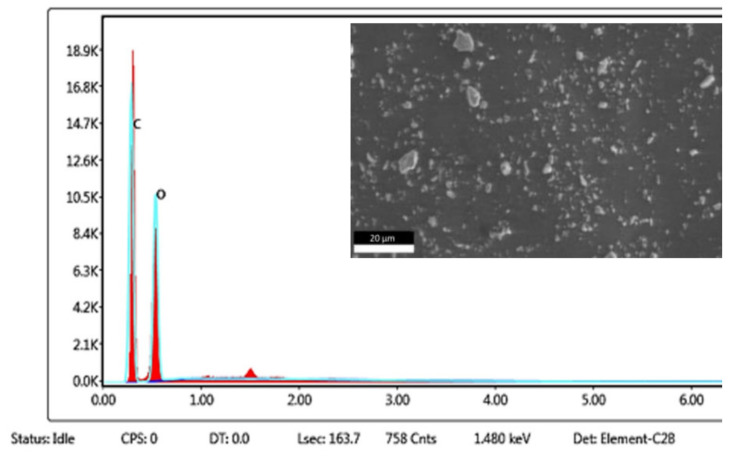
SEM-energy dispersive X-ray (EDX) analysis of the lignin powder.

**Figure 3 polymers-13-01109-f003:**
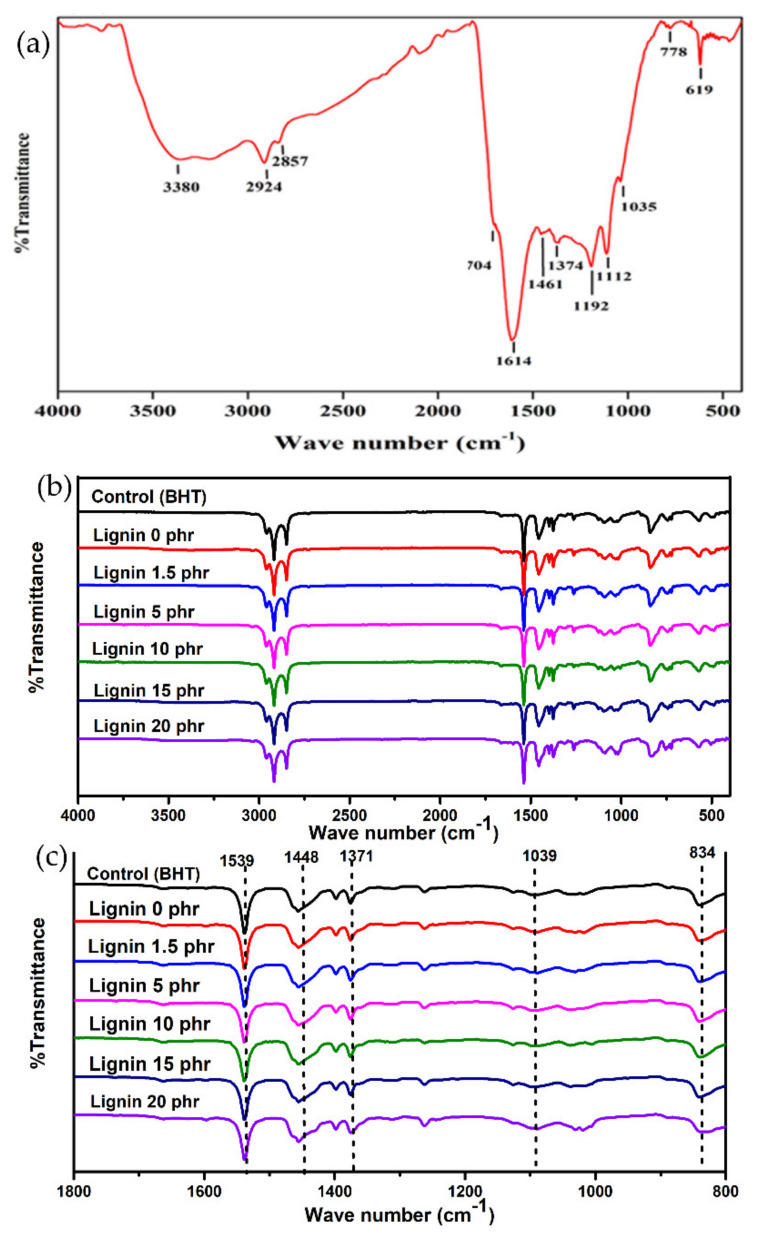
FTIR spectra of (**a**) the pure lignin, and natural rubber composites (**b**) over the wavenumbers from 4000 to 500 cm^−1^ and (**c**) over the wavenumbers from 1800 to 800 cm^−1^.

**Figure 4 polymers-13-01109-f004:**
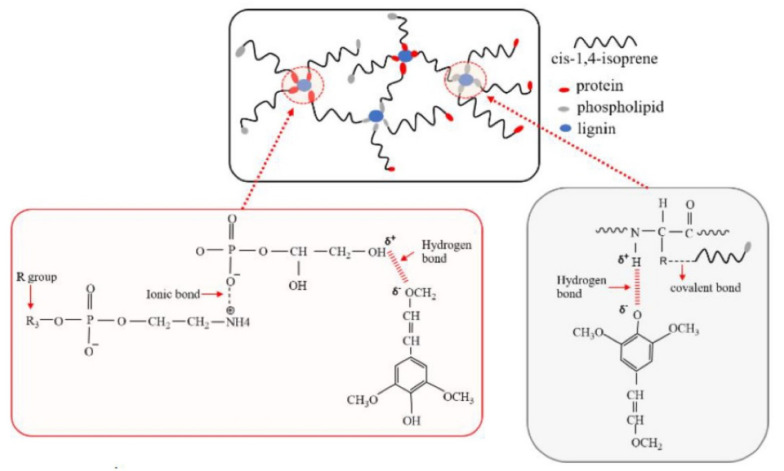
The proposed illustration of bonding mechanism of lignin on non-rubber.

**Figure 5 polymers-13-01109-f005:**
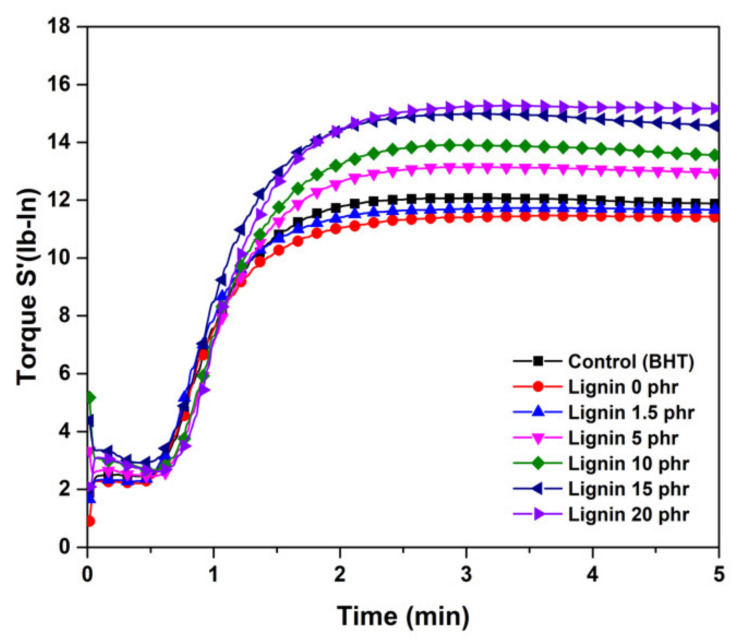
Curing curves of the NR/lignin compounds.

**Figure 6 polymers-13-01109-f006:**
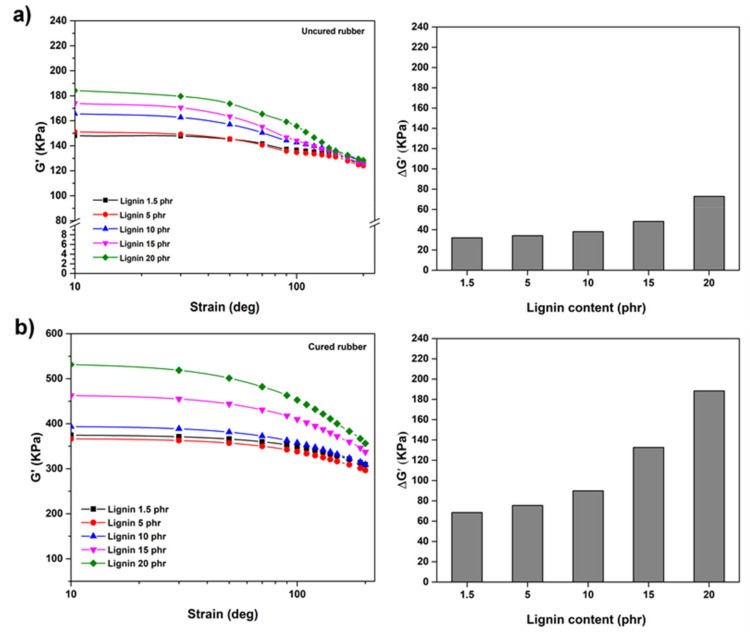
Shear modulus (G’) by strain for uncured (**a**) and cured (**b**) compounds.

**Figure 7 polymers-13-01109-f007:**
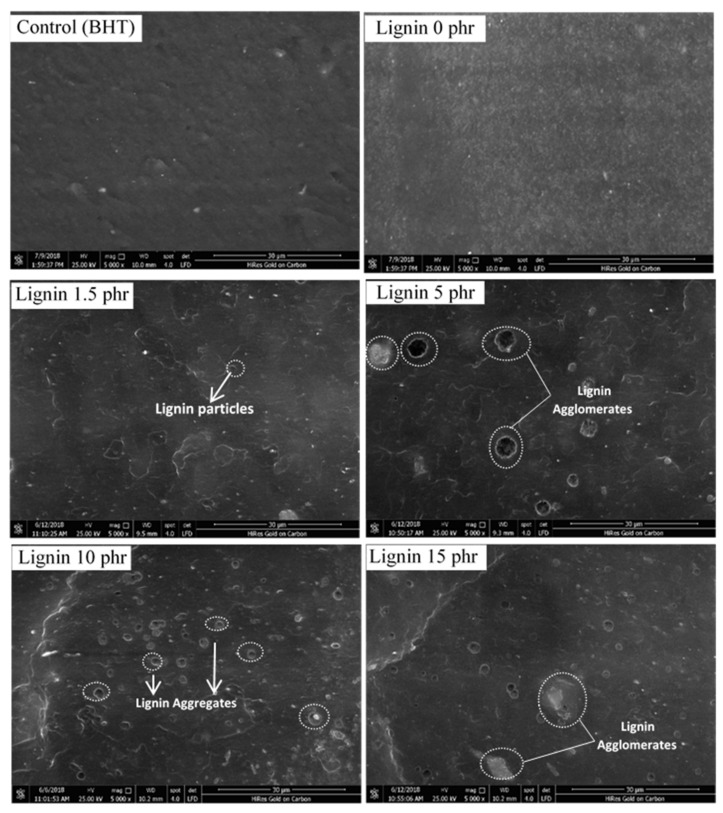
SEM micrographs of NR and lignin-filled rubber composites.

**Figure 8 polymers-13-01109-f008:**
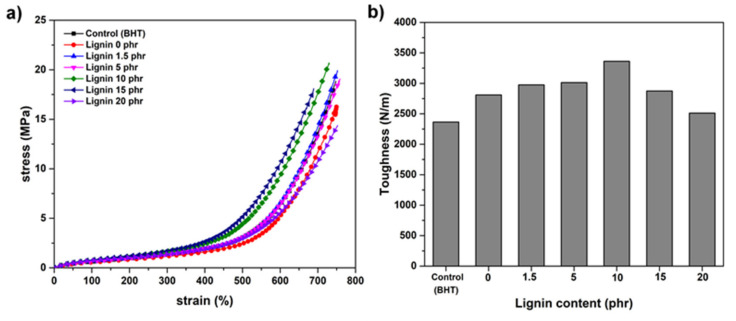
Typical tensile stress–strain curves (**a**) and toughness (**b**) for the rubber composites.

**Figure 9 polymers-13-01109-f009:**
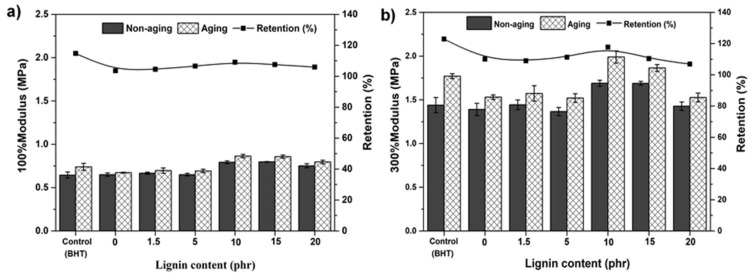
The 100% (**a**) and 300% (**b**) modulus of NR/lignin vulcanizates with different lignin contents.

**Figure 10 polymers-13-01109-f010:**
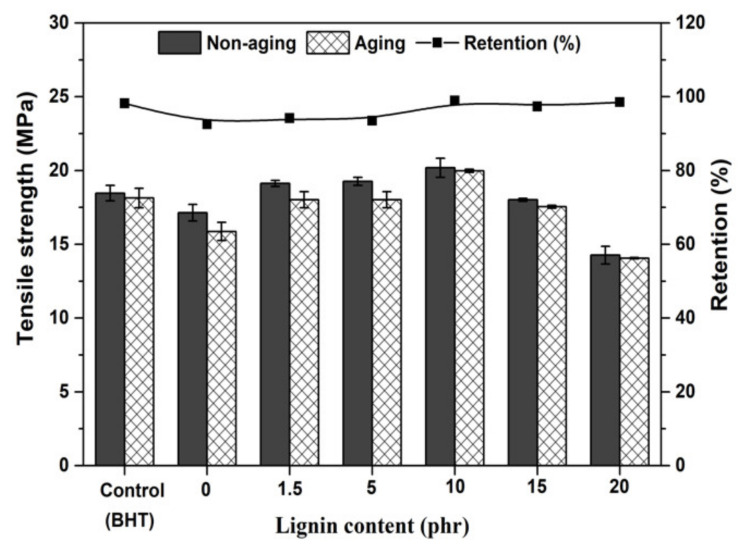
Tensile strength of NR/lignin vulcanizates with different lignin contents.

**Figure 11 polymers-13-01109-f011:**
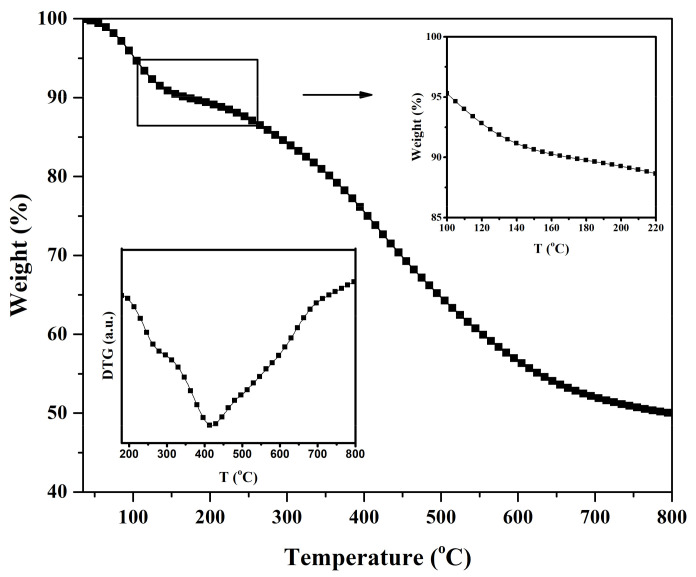
TGA and DTG responses of lignin to heating at 10 °C/min.

**Figure 12 polymers-13-01109-f012:**
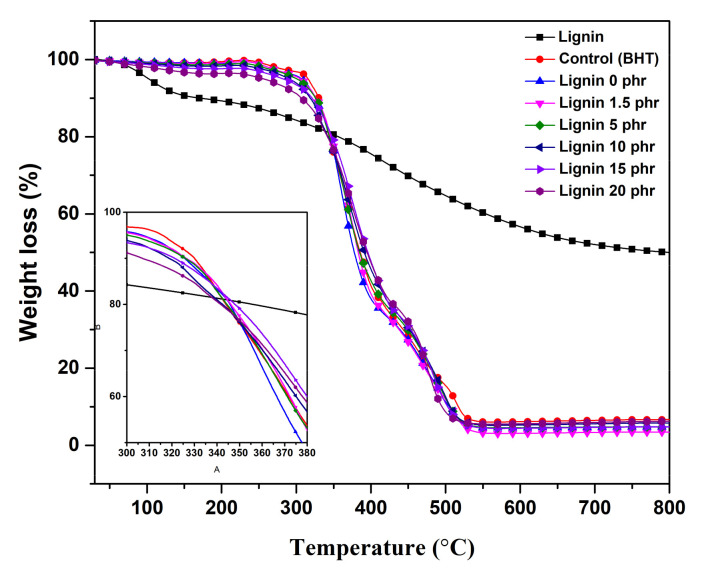
TGA thermograms of lignin and natural rubber/lignin composites.

**Figure 13 polymers-13-01109-f013:**
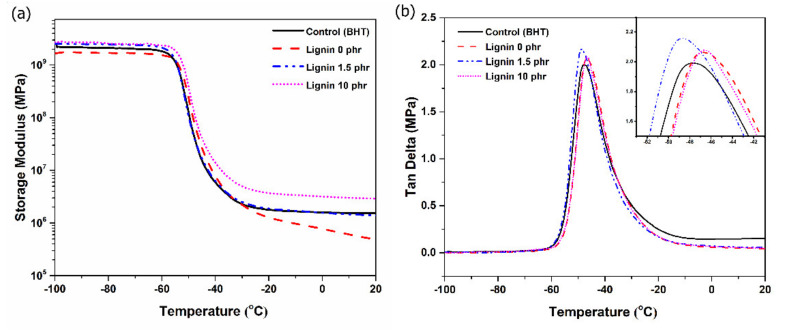
Storage modulus (E ‘) (**a**) and Tan δ (**b**) of NR and lignin-filled NR composites.

**Table 1 polymers-13-01109-t001:** Compounding formulation (part per hundred of rubber, phr) of NR/Butylated Hydroxyl Toluene (BHT)/Lignin in this work.

Sample Code	NR/BHT/Lignin
Unfilled	100/0/0
Control (BHT)	100/1.5/0
Lignin 1.5	100/0/1.5
Lignin 5	100/0/5
Lignin 10	100/0/10
Lignin 15	100/0/15
Lignin 20	100/0/20

**Table 2 polymers-13-01109-t002:** Potential peak assignments in the FTIR spectra.

Band Position (cm^−1^)	Assignment
3396–3406	O–H Stretching
2923	C–H Stretching
2849	C–H Stretching
1704	C=O Stretching Unconjugated
1614	C=O Stretching Conjugated
1461	CH_2_ Deformation stretching
1374	C–O stretching of the syringyl ring
1192	C–C and C–O stretching vibration of guaiacyl ring
1035	C–H stretching of Aromatic in guaiacyl ring and C–O from primary alcohol
1112	C–H stretching in Aromatic deformation in the syringyl ring
778 and 619	C–H out of plane in positions 2, 5 and 6 (G units)

**Table 3 polymers-13-01109-t003:** The Mooney viscosities of rubber composites.

Lignin Content	Mooney Viscosity (ML 1 + 4 (100 °C))
Control (BHT)	37.5
Lignin 0 phr	40.9
Lignin 1.5 phr	39.1
Lignin 5 phr	38.7
Lignin 10 phr	35.2
Lignin 15 phr	35.1
Lignin 20 phr	34.9

**Table 4 polymers-13-01109-t004:** Curing characteristic parameters of the rubber compounds with various lignin loadings ^a^.

Formula	ts2 (min)	tc90 (min)	ML (dN m)	MH (dN m)	ΔM (dN m)
Control (BHT)	0.49	1.41	2.36	12.31	9.95
NR 0lignin	0.45	1.41	2.17	11.78	9.61
NR 1.5lignin	0.51	1.51	2.36	12.84	10.48
NR 5lignin	0.55	1.57	2.33	13.31	10.98
NR 10lignin	1.02	2.13	2.44	14.29	11.85
NR 15lignin	1.10	2.37	2.53	14.55	12.02
NR 20lignin	0.52	2.06	2.62	15.27	12.65

^a^ t_S_2: scorch time; tc90: optimum cure time; ML: the minimum torque; MH: the maximum torque; ΔM: the difference between maximum torque and minimum torque.

**Table 5 polymers-13-01109-t005:** Thermal degradation of NR and lignin-filled NR composites.

Lignin Content	Decomposition Temperature (T_50_) (°C)
Control (1.5 BHT)	377.67
Lignin 0 phr	357.90
Lignin 1.5 phr	383.00
Lignin 5 phr	384.83
Lignin 10 phr	390.83
Lignin 15 phr	395.83
Lignin 20 phr	394.83

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
