# Peer review of "The Characteristics of Natural Rubber Composites with Klason Lignin as a Green Reinforcing Filler: Thermal Stability, Mechanical and Dynamical Properties"

_polymers, 2021, doi:10.3390/polym13071109_

Round 1
Reviewer 1 Report
The manuscript entitled “Rubberwood Klason lignin as an alternative green reinforcing filler in natural rubber vulcanizates: thermal stability and mechanical properties” investigates the effect of Klason lignin filler on the thermal stability and properties of natural rubber composites. The results showed that Klason lignin filler could be used as green antioxidant and alternative reinforcing filler for high-performance natural rubber biocomposites. The reported work is within the scope of Polymers, and the results were supported by the experimental data. However, several points should be addressed before publication.
1. The following papers should be included to show your grasp of the achievements in the field of lignin such as Journal of Bioresources and Bioproducts, 2020, 5(2):79–95; Journal of Bioresources and Bioproducts, 2020, 5(3):163-179, etc.
2. There are many format errors on the current manuscript. The author should be carefully revised it in the revision. For example, “1374 cm-1” should be “1374 cm-1”; Table 1 had some different formats on NR/BHT/Lignin section.
3. The EDX data in Figure 2 is not clear and distorted. Please provide the clear version in the revision. Also, many other figures in the current version have been distorted. Please revise it carefully in the revised manuscript.
4. The addition of 1.5phr lignin showed lower minimum torque, maximum torque and delta torque. Can the author explain the phenomenon and add it into the revised manuscript?
5. Figure 7 presented the poor dispersion and interaction of lignin filler to natural rubber matrix in Lignin 10 phr sample. In Figure 9, the 100% modulus and tensile strength of the Lignin 10 phr sample were highest. However, in Figure 7, the SEM image of Lignin 15 phr sample displayed better dispersion than Lignin 10 phr sample. The author should be re-discussed these parts again in the revision.
6. The caption of Figure 9 is wrong. Please check it.
Author Response
Following list is a response to the questions and comments of the
Reviewer # 1. Any revisions within the manuscript were highlighted in yellow color.
1-The following papers should be included to show your grasp of the achievements in the field of lignin such as Journal of Bioresources and Bioproducts ,2020,5(2):79-95; Journal of Bioresources ,2020,5(3):163-179, etc.
Based on the suggestions of the reviewer below two publications from the current researches were added and cited in the manuscript accordingly.
1-Xia, Z.; Li, J.; Zhang, J.; Zhang, X.; Zheng, X.; Zhang, J. Processing and
Valorization of Cellulose, Lignin and Lignocellulose Using Ionic Liquids.
- Bioresour. and Bioprod. 2020, 5(2): 79-95.
2-Lia, H.; Lianga,Y.; Lia,P.; Heb, C. Conversion of biomass lignin to high-
value polyurethane: A review.2020, J. Bioresour. and Bioprod. 5(3): 163-
2-There are many format errors manuscript, The author should be carefully revised it in the revision. For example, “1374 cm-1” should be “1374 cm-1”; Table 1 had some different formats on NR/BHT/Lignin section.
Complete manuscript was proofread and format errors including above inconsistencies were corrected within the text.
3-The EDX data in Figure 2 is not clear and distorted. Please provide the clear version in the revision. Also, many other figures in the current version have been distorted. Please revise it carefully in the revised manuscript.The addition of 1.5phr lignin showed lower minimum torque, maximum torque and delta torque. Can the author explain the phenomenon and add it into the revised manuscript?
Manuscript was revised and below explanation was added to make it rather clearer. Also distorted figures were revised to their new format.
The minimum torque, maximum torque, and delta torque of the 1.5 phr lignin‐filled NR composites were insignificantly lower than those of the control BHT as illustrated in Figure 5. The trend of maximum torque revealed that 1.5 phr of lignin loading did not contribute to the in-creased shear modulus of the natural rubber composite during vulcanization.
4- Figure 7 presented the poor dispersion and interaction of lignin filler to natural rubber matrix in Lignin 10 phr sample. In Figure 9, the 100% modulus and tensile strength of the lignin 10 phr sample were highest, However, in Figure 7, the SEM image of Lignin 15 phr sample displayed better dispersion than Lignin 10 phr sample, The author should be re-discussed these parts again in the revision.
Lignin content at 10phr gave homogeneous filler dispersion with small particle sizes and the fracture surface had small holes. Comparing to 15 phr lignin filled-rubber, overall content at 10phr showed less agglomerated lignin than 15 phr . Such finding indicates fully dispersion of 10 phr lignin in the rubber matrix and interfacial adhesive between lignin and rubber, matching the improved with the crosslink density and mechanical properties (100%, 300% modulus, toughness and tensile strength). This statement was included in the manuscript.
5- The caption of Figure 9 is wrong. Please check it.
Inaccuracies in Figure 9 was corrected.

Reviewer 2 Report
I read an interesting and comprehensive research work entitled ‘Rubberwood Klason lignin as an alternative green reinforcing filler in natural rubber vulcanizates: thermal stability and mechanical properties’. The concept of the manuscript completely fits and is suitable to publish in Polymers Journal. This manuscript is generally well written and clearly presented however still needs to address many comments and thus require major revision.
1) English and grammar mistakes are present. The author should check the manuscript by native English Speaker to improve the quality of the manuscript.
2) Title should be modified in a precise way. Abstract section should be rewritten and look very general and not informative. In abstract authors should give the details of results and mention the importance of lignin as reinforcing filler briefly.
3) Provide a nice graphical abstract representing the overview of the MS with key highlights.
4) This manuscript is lacking to cite the recent review of literature; hardly few papers have been cited of the year 2018-2020.
5) In the introduction section, write the novelty of the work and the problem statement clearly. Authors should discuss some recent applications of lignin Viz. NPs synthesis and biopolymers production pl refer and cite “International journal of biological macromolecules 128, 391-40, 2019”; “Bioresource Technology Volume 325, April 2021, 124685”. The detailed discussion about the novelty, significance of your research work and research gap relative to the literature is essential.
6) Why authors choose klason lignin need substantial discussion.
7) For analytical studies give all operational details, and provide suitable reference. I hope authors will pay attention on this during revisions stage.
8) Statistical analysis of the results should be provided in the materials and methods section. It's important for all experimental work Report these values in the results and discussion.
9) More discussion of the FTIR results with literature is essential. For line n o 211-213 give suitable reference. Similarly throughout the manuscript authors just discussed their results but not compared or discussed with the literature which is surprising. Major changes are expected during the revision stage.
10) In figure and table always give full form of abbreviation. In addition, for figure and table caption give all details.
11) Write the practical applications and future research perspectives and challenges by adding a new section before conclusions.
12) What are the limitation to use this methodology for commercial application?
13) The conclusion of the study is not discussed with the specific output obtained from the study, it could be modified with precise outcomes with a take home message.
Author Response
Following list is a response to the questions and comments of the Reviewer # 2. Any revisions within the manuscript were highlighted in yellow color.
- English and grammar mistakes are present. The author should check the manuscript by native English Speaker to improve the quality of the manuscript.
Complete manuscript was proofread to correct English language and grammatical errors by highlighting in yellow color and revised to its new enhanced format.
- Title should be modified in a precise way. Abstract section should be rewritten and book very general and not informative. In abstract authors should give the details of results and mention the importance of lignin as reinforcing filler briefly
The title of the article was changed to “The Characteristics of natural rubber composites having Klason lignin as a green reinforcing filler: Thermal stability, mechanical and dynamical properties”
Also Abstract was revised to its new format as suggested by the reviewer.
- Provide a nice graphical abstract representing the overview of the MS with key highlights.
A new “Graphical Abstract” as illustrated below was included in JPEG file.
- This manuscript is lacking to cite the recent review of literature; hardly few papers have been cited of the year 2018-2020.
Below six current articles were added in the manuscript and cited at the proper places in the text accordingly. We believe overall quality of our study has been enhanced with addition of the below articles.
1-Xia, Z.; Li, J.; Zhang, J.; Zhang, X.; Zheng, X.; Zhang, J. Processing and Valorization of Cellulose, Lignin and Lignocellulose Using Ionic Liquids. J. Bioresour. and Bioprod. 2020, 5(2): 79-95.
2-Barana, D., Ali, S. D., Salanti, A., Orlandi, M., Castellani, L., Hanel, T. Influence of lignin features
on thermal stability and mechanical properties of natural rubber compounds. ACS Sustain. Chem. Eng. 2016, 4, 5258–5267.
3-Wang, H., Liu, W., Huang, J., Yang, D., and Qiu, X. Bioinspired engineering towards tailoring
advanced lignin/rubber elastomers. Polym. 2018,10,1033.
4-Lang, J. M., Shrestha, U. M., and Dadmun, M. The effect of plant source on the properties of lignin-
based polyurethanes. Front. Energy Res. 2018, 6:4.
5-Lia, H.; Lianga,Y.; Lia,P.; Heb, C. Conversion of biomass lignin to high-value polyurethane: A
review. J. Bioresour. and Bioprod. 2020, 5: 163-179.
6-Ganesh Saratale, R.; Cho, S.-K.; Dattatraya Saratale, G.; Kadam, A.A.; Ghodake, G.S.; Kumar, M.; Naresh Bharagava, R.; Kumar, G.; Su Kim, D.; Mulla, S.I. A comprehensive overview and recent advances on polyhydroxyalkanoates (PHA) production using various organic waste streams. Bioresour. Technol. 2021, 325, 124685.
- In the introduction section, write the novelty of the work and the problem statement clearly Authors should discuss some resent applications of lignin Viz NPs synthesis and biopolymers production pl refer and cite “International journal of biological macromolecules 128,391-40,2019”; “ Bioresource Technology Volume 325,April 2021,124685 “.The detailed discussion about the novelty ,significance of your work and research gap relative to the literature is essential.
Certain parts of the Introduction was revised to its new format and below suggested article was added into the work and cited accordingly.
Ganesh Saratale, R.; Cho, S.-K.; Dattatraya Saratale, G.; Kadam, A.A.; Ghodake, G.S.; Kumar, M.; Naresh Bharagava, R.; Kumar, G.; Su Kim, D.; Mulla, S.I. A comprehensive overview and recent advances on polyhydroxyalkanoates (PHA) production using various organic waste streams. Bioresour. Technol. 2021, 325, 124685.
- Why authors choose klason lignin need substantial discussion.
The main reason Klason lignin was considered in this work it is fact that extracted Klason lignin showed high purity and content of carbon, oxygen with very high temperature decomposition (424 0C). Lignin filled-NR can also be considered to be used as an alternative green bio-composite with thermal aging resistant and low rolling resistant for tire industry.
- For analytical studies give all operational details, and provide Suitable reference. I hope authors will pay attention on this during……
Operational details were added and highlighted in yellow color in the manuscript along with addition of proper reference.
- Statistical analysis of the results should be provided in the materials and methods section. It’s important for all experimental work Report these values in the results and discussion.
It was not to objective of this study to run large amount of sample size at this point but in the next phase. We don’t believe that statistical analysis will be meaningful at this level.Therefore we just want to explore initial data which will be used and expanded more comprehensive in the next phase of the study. Statistical analysis mainly ANOVA will be employed during the second phase of the work.
- More discussion of the FTIR results with literature is essential. For Line n o 211-213 give suitable reference. Similarly throughout the Manuscript authors just discussed their results but not compared or Discussed with the literature which is surprising. Major changes are Expected during the revision stage.
Two new references as listed related to FTIR analysis were added and discussed in the proper places of the manuscript and highlighted in yellow color.
1-Hu J.; Xiao R.; Shen D.; Zhang H. Structural analysis of lignin residue from black
liquor and its thermal performance in thermogravimetric-Fourier transform infrared
spectroscopy. Bioresource Technol. 2013, 128:633–639.
2-Pandey, K. A study of chemical structure of soft and hard wood and wood polymers
by FTIR spectroscopy. J. Appl. Polym. Sci., 1999, 71, 1969-1975.
- In figure and table always give full form of abbreviation. In addition, for figure and table caption give all details.
Figures and Tables were revised as suggested to their new configurations.
- Write the practical applications and future research perspectives and challenges by adding a new section before conclusions.
A further research objective would include the method treatment of Klason lignin by modifying process or using coupling agent in order to increase interaction with natural rubber matrix. The modifying and treatment would also give the opportunity of using klason lignin for different applications once combination of coupling and modification, higher filler-rubber interaction will be gained.
- What are the limitation to use this methodology for commercial
application?
-This limitation of lignin extraction for commercial are only strong of sulfuric treatment , however under controlling system with automatic and close system this method is unexpansive and give pure of lignin and high carbon content. This lignin can be use as substrate for many industry.
- The conclusion of the study is not discussed with the specific
output obtained from the study, it could be modified with precise
outcomes with a take home message.
-In this study, a detailed characterization of structural Klason lignin and thermal degradation of lignin was accomplished. The antioxidant properties of Klason lignin as a reinforcing filler in natural rubber composites, was investigated within the perspective of this study. It was determined that Klason lignin from rubber wood composed both guaiacyl and syringyl structure with high content of carbon and oxygen which can be interacted with non-rubber content of natural rubber. It can be concluded that, lignin increased the scorch time improving processibility, and also increased delta torque, modulus levels of 100% and 300% and tensile strength of vulcanized rubber. The antioxidant properties of Klason lignin were found as superior to the commercial antioxidant additive BHT. It appears that the 10phr lignin loading was near optimal resulting in the highest mechanical properties and aging resistance of composite samples. The SEM analyzing showed a homogeneous distribution of lignin particles in the rubber matrix, which improved the toughness property of the samples. DMA indicated that Klason lignin / NR had high wet grip properties revealing that the samples had low rolling resistance, which could potentially make them for green tire products. Moreover, the use of lignin as a filler not only facilitated the processing and also provided an environmentally friendly value-added
